# OpenReview forum: "DexBench: Benchmarking LLMs for Personalized Decision Making in Diabetes Management"
_ICLR.cc/2026/Conference — Submitted to ICLR 2026_

### Official Review · Reviewer_FnwP · 2025-10-30

**Soundness:** 3
**Presentation:** 3
**Contribution:** 2
**Rating:** 4
**Confidence:** 4

**Summary:**

This paper presents DexBench, a benchmark designed to evaluate large language models on real-world, patient-facing diabetes management tasks. Built from data of 15,000 individuals across type 1, type 2, and prediabetes populations, it generates 360,600 contextualized questions covering seven representative tasks derived from CGM and behavioral data. The authors also design a five-dimensional evaluation framework (accuracy, groundedness, safety, clarity, and actionability) and assess eight diverse LLMs. The study offers a framework for assessment of LLM performance in diabetes-related contexts.

**Strengths:**

1. The study constructs a large, diverse, and multimodal patient-facing diabetes dataset, covering comprehensive real-world management tasks and complementing existing benchmark efforts.

2. It systematically evaluates multiple LLMs, discusses their strengths and limitations, and provides some useful insights for improving diabetes-specific model capabilities.

**Weaknesses:**

1. As acknowledged by the authors, the dataset lacks detailed demographic information (e.g., age) and omits important variables such as insulin use and medication data. In addition, it relies heavily on wearable and self-reported inputs, which may be sparse, noisy, and limit the robustness and representativeness of the benchmark.

2. The evaluation is primarily based on model-generated responses, which may introduce bias. Incorporating domain-specific diabetes knowledge or clinical expertise into the evaluation process could improve the rigor and reliability of model assessment.

**Questions:**

1. Could the authors clarify the dataset source in detail—specifically, how the real-world data were collected and whether the reported cohort of 15,000 individuals includes any synthetically generated data? ps. The prediabetes/health and wellness group may not be a true diabetes population; clarification in terminology would improve precision.

2. Although the paper notes missing demographic (e.g., age, sex) and treatment variables (insulin, medications), , the benchmark distinguishes adults and adolescents in Task 2. Could the authors provide basic dataset statistics, such as the proportion of adults vs. adolescents and the completeness rate of self-logged data? This information would help assess the representativeness and generalizability of the bench.

3. Lines 190-191 mention that a human expert manually confirms the quality of generated questions. Could the authors elaborate on the review process and criteria used to judge question quality?

4. The related work section states that previous diabetes benchmarks are clinician-facing, but some prior efforts are not strictly clinician-oriented. The authors may consider clarifying or citing those examples for completeness.

5. Given that most evaluations rely on model-based scoring, how do the authors ensure scoring fairness, especially when the scoring model itself may not be the strongest performer? What were the considerations in selecting the scoring model? In addition, for binary criteria, could a graded or probabilistic scoring scheme provide a more nuanced assessment?

6. Lines 713-715 describe accuracy as “agreement with ground-truth values within ±2 mg/dL, with no calculation errors permitted.” Could the authors clarify this definition? There seems to be a potential inconsistency. Besides, not all metrics (e.g., TIR) are expressed in mg/dL.

7. Regarding the clarity metric (Flesch-Kincaid Grade Level), could the authors explain how this measure was implemented and validated for health communication contexts?

8. The reported benchmark scores are relatively high. Does this suggest that the current benchmark may not pose sufficient challenge to newer models? How might future updates maintain or enhance its discriminative ability?

9. The paper mentions that it plans to extend DexBench to other health domains. Could the authors elaborate on how the framework could be adapted to chronic conditions such as hypertension or obesity? Have any preliminary steps been taken in that direction?

---

> ### Author Response · Authors · 2025-11-20
> **Response to Reviewer FnwP**
>
> Thank you for the helpful review, and, in particular, for the acknowledgement of the *scale, diversity, and multimodal design of DexBench*, as well as the *systematic evaluation and analysis of LLM strengths and limitations in the context of diabetes management*. We address below the outstanding questions.
>
> > **Weakness 1.As acknowledged by the authors, the dataset lacks detailed demographic information (e.g., age) and omits important variables such as insulin use and medication data. In addition, it relies heavily on wearable and self-reported inputs, which may be sparse, noisy, and limit the robustness and representativeness of the benchmark.**
>
> We appreciate this important observation. We agree that the absence of demographic and condition-specific information (e.g., age, insulin use, medication data) may limit certain forms of reasoning analysis. **This limitation is explicitly acknowledged in Section 5 and further in Appendix A.3**, where we note that such variables are not available in the current dataset. We also recognize that wearable and self-reported data can be sparse or noisy, as noted in Section 5. In future work, we plan to collect additional data to incorporate richer demographic and clinical features to **enhance task realism, improve robustness, and enable more personalized model evaluation across diverse user contexts**.
>
> > **Weakness 2.The evaluation is primarily based on model-generated responses, which may introduce bias. Incorporating domain-specific diabetes knowledge or clinical expertise into the evaluation process could improve the rigor and reliability of model assessment.**
>
> We thank the reviewer for this insightful comment.
> For evaluation, we utilized a **deterministic grader (Gemini 2.5 Pro; temperature/top-p = 0), complemented by human expert verification on random subsets**.
> These experts are **domain specialists in diabetes management**, including junior experts with approximately two years of experience and senior experts with five or more years of experience working deeply with diabetes populations and multimodal diabetes data. They hold PhDs and work at a leading diabetes technology company, ensuring familiarity with both clinical reasoning and data-driven decision-making. Their role was to **verify LLM grader outputs and confirm overall quality** by manually reviewing random subsets of generated questions and graded responses.
> To improve confidence and reliability, **we implemented a meta-evaluation, validating model agreement with human experts through targeted expert grading**. We sampled one random question/answer pair per task, per cohort, and per model **(168 pairs;840 metrics)** and had one senior and one junior expert independently assign binary (0/1) scores to each metric for each response. We then **computed Cohen's κ to quantify inter-rater reliability**.
>
> | Comparison         | Model–AvgExpert | Model–SrExpert | Model–JrExpert | SrExpert–JrExpert |
> |--------------------|----------------:|----------------:|----------------:|------------------:|
> | Overall κ          | 0.79            | 0.92            | 0.67            | 0.71              |
>
> These results provide **quantitative evidence of strong model alignment with human experts (κ = 0.79), exceeding the agreement observed between experts themselves (κ = 0.71)**. We will incorporate these findings, along with additional alignment analyses, into the revised paper.

---

> > ### Author Response · Authors · 2025-11-20
> > **Response to Reviewer FnwP**
> >
> > > **Question 1.Could the authors clarify the dataset source in detail—specifically, how the real-world data were collected and whether the reported cohort of 15,000 individuals includes any synthetically generated data? ps. The prediabetes/health and wellness group may not be a true diabetes population; clarification in terminology would improve precision.**
> >
> > We thank the reviewer for this helpful question. The **real-world data** used in DexBench were contributed by **15,000 unique users (5,000 per cohort)**. Each user contributed **30 days of consecutive data**, where the most complete 30 days were selected from the user between January and June 2025. The glucose traces were collected from **continuous glucose monitoring (CGM) devices**, specifically Dexcom's G7 device for the T1D cohort and Dexcom's Stelo device for the T2D and HW cohorts. The **CGM data is recorded continuously every 5 minutes**. The **behavioral data came from user self-reported logs** (e.g., about what meal they ate and when), and the exercise, sleep and daily activity summaries came from wearable smart watches. We have clarified this information in the revised manuscript.
> >
> > Importantly, **the primary DexBench cohort consists of these 15,000 real users; no synthetic individuals were included in the main dataset**. This real-world data was used for Tasks 3-7, but not for Task 1 (Glucose Math), which required complete CGM traces. Because releasing such granular data is not possible due to legal and privacy concerns, **Task 1 uses physiologically realistic, privacy-preserving CGM traces generated via GlucoSynth [A] (Appendix A.3)**, a widely recognized tool for synthetic CGM generation. Using [A], we generated synthetic glucose traces of 15,000 individuals, with 5,000 from each cohort (type 1, type 2, prediabetes/general health) similar to the main dataset of real users. We believe this approach is appropriate because Task 1 is focused on evaluating understanding of glucose dynamics and mathematical reasoning, rather than behavioral associations or personalized context as in other tasks. **The synthetic cohort is entirely independent of the real users and was generated solely to maintain consistency in sample size and task structure while enabling open data release.** Thus, we have a main dataset of real users (n = 15,000) used for Tasks 3-7, and a synthetic dataset (n = 15,000) used for Task 1. We will clarify this distinction more explicitly in the revised manuscript.
> >
> > We also agree that the term **“prediabetes/general health”** could be described more precisely. This cohort includes individuals diagnosed with prediabetes as well as those without diabetes, **grouped together to represent a healthier population segment for comparison**. We included this group **to capture a broader spectrum of glucose dynamics—ranging from normal to unregulated patterns—and to test model reasoning across less variable, preventive-care contexts.** Individuals in this cohort may also have family histories or risk factors for diabetes, making them relevant to early-stage behavioral reasoning tasks. We will clarify this terminology and rationale in the revised version.

---

> > > ### Author Response · Authors · 2025-11-20
> > > **Response to Reviewer FnwP**
> > >
> > > > **Question 2.Although the paper notes missing demographic (e.g., age, sex) and treatment variables (insulin, medications), , the benchmark distinguishes adults and adolescents in Task 2. Could the authors provide basic dataset statistics, such as the proportion of adults vs. adolescents and the completeness rate of self-logged data? This information would help assess the representativeness and generalizability of the bench.**
> > >
> > > We thank the reviewer for this important question. The current dataset does not include demographic attributes, including age or the proportion of adults versus adolescents. To clarify, **Task 2 (Education)** does not rely on user data for question generation or answering. Instead, it is a **conversational task** designed to simulate **educational interactions tailored for two groups—adults and adolescents—** to ensure that DexBench captures reasoning and communication patterns relevant to both populations. We will clarify this distinction in the revised version.
> > >
> > > We report the **completeness rate of self-logged data to help evaluate the representativeness and generalizability of the benchmark**. Completeness was defined as the proportion of days in which an individual logged at least one instance of a given behavior, yielding a value between 0 and 1. This simple formulation provides a transparent and comparable measure of logging frequency across cohorts and behaviors.
> > >
> > > | **Behavior** | **HW** | **T1D** | **T2D** |
> > > |---------------|-------:|--------:|--------:|
> > > | Sleep         | 0.94   | 0.00    | 0.21    |
> > > | Exercise      | 1.00   | 1.00    | 0.95    |
> > > | Meals         | 0.25   | 0.78    | 0.36    |
> > >
> > > These results suggest that **data completeness was generally high, although logging frequency varied across behaviors and cohorts**. We will add in completeness details and analysis in the revised manuscript.
> > >
> > > > **Question 3.Lines 190-191 mention that a human expert manually confirms the quality of generated questions. Could the authors elaborate on the review process and criteria used to judge question quality?**
> > >
> > > We thank the reviewer for this helpful question. **Domain experts manually reviewed a representative sample of generated questions to ensure overall quality and coherence.**
> > > These experts are **domain specialists in diabetes management**, including **junior experts with approximately two years of experience** and **senior experts with five or more years of experience** working deeply with diabetes populations and multimodal diabetes data. They hold PhDs and work at a leading diabetes technology company, ensuring familiarity with both clinical reasoning and data-driven decision-making.
> > > **These domain experts assessed whether each question was fluent, relevant, original, difficult, and answerable** (Table 5). This process ensured that the questions were **well-formed and suitable for evaluating reasoning quality**. Because questions themselves cannot be “incorrect” in the same sense as model answers, we adopted this **expert verification approach as sufficient for the first version** of the benchmark, balancing rigor with scalability.
> > >
> > > > **Question 4.The related work section states that previous diabetes benchmarks are clinician-facing, but some prior efforts are not strictly clinician-oriented. The authors may consider clarifying or citing those examples for completeness.**
> > >
> > > We thank the reviewer for this thoughtful observation. In Section 4 (Related Work), we discuss prior diabetes-related benchmarks such as Xie & Wang (2020) [B] and Healey et al. (2025) [C], which are primarily clinician-facing, while LLM-CGM (Healey & Kohane, 2024) [D] is in fact patient-facing, focusing on CGM interpretation for individual users. We appreciate the reviewer’s point that **not all prior efforts in this category are strictly clinician-oriented**. We will clarify this distinction and adjust our phrasing in the revised version for completeness.

---

> > > > ### Author Response · Authors · 2025-11-20
> > > > **Response to Reviewer FnwP**
> > > >
> > > > > **Question 5.Given that most evaluations rely on model-based scoring, how do the authors ensure scoring fairness, especially when the scoring model itself may not be the strongest performer? What were the considerations in selecting the scoring model? In addition, for binary criteria, could a graded or probabilistic scoring scheme provide a more nuanced assessment?**
> > > >
> > > > We thank the reviewer for this helpful question.
> > > > For evaluation, we utilized a **deterministic grader (Gemini 2.5 Pro; temperature/top-p = 0), complemented by human expert verification on random subsets**.
> > > > These experts are **domain specialists in diabetes management**, including junior experts with approximately two years of experience and senior experts with five or more years of experience working deeply with diabetes populations and multimodal diabetes data. They hold PhDs and work at a leading diabetes technology company, ensuring familiarity with both clinical reasoning and data-driven decision-making. Their role was to **verify LLM grader outputs and confirm overall quality** by manually reviewing random subsets of generated questions and graded responses.
> > > > To improve confidence and reliability, **we implemented a meta-evaluation, validating model agreement with human experts through targeted expert grading**. We sampled one random question/answer pair per task, per cohort, and per model **(168 pairs;840 metrics)** and had one senior and one junior expert independently assign binary (0/1) scores to each metric for each response. We then **computed Cohen's κ to quantify inter-rater reliability**.
> > > >
> > > > | Comparison         | Model–AvgExpert | Model–SrExpert | Model–JrExpert | SrExpert–JrExpert |
> > > > |--------------------|----------------:|----------------:|----------------:|------------------:|
> > > > | Overall κ          | 0.79            | 0.92            | 0.67            | 0.71              |
> > > >
> > > > These results provide **quantitative evidence of strong model alignment with human experts (κ = 0.79), exceeding the agreement observed between experts themselves (κ = 0.71)**. We will incorporate these findings, along with additional alignment analyses, into the revised paper.
> > > >
> > > > The **scoring model** was chosen to be Gemini 2.5 Pro, **the strongest Gemini model** at the time, set with temperature/top-p set to 0.0 for deterministic scoring. This choice was based on its **demonstrated performance in prior evaluation case studies, consistency under deterministic decoding, and practical considerations** such as availability and seamless integration under existing enterprise agreements.
> > > >
> > > > We intentionally use binary (0/1) metrics because DexBench evaluates safety-critical, patient-facing reasoning. **In realistic deployment, an output that is even partially unsafe, ungrounded, or factually incorrect must be treated as a failure**. For example, a response containing any unsafe medical advice should receive a safety score of 0. **Each criterion we evaluate (accuracy, safety, groundedness, clarity, and actionability) is governed by an explicit rubric that defines what constitutes a 0 versus 1 for each task** (Appendix A.5), ensuring consistent and reproducible judgments.
> > > >
> > > > This binary formulation aligns with how **clinical decision-support systems are commonly validated**, and in prior medical benchmarking work e.g., [C]. It also simplifies evaluation across thousands of questions and metrics (1,803,000 total metrics for all questions), enabling **scalable and unambiguous benchmarking**.
> > > >
> > > > We agree that incorporating finer-grained or weighted scoring could provide additional nuance, and we plan to explore this direction in future work. However, for the initial release, **binary grading offered the most practical and interpretable foundation** for establishing reliability in a large, safety-critical benchmark.
> > > >
> > > > > **Question 6.Lines 713-715 describe accuracy as “agreement with ground-truth values within ±2 mg/dL, with no calculation errors permitted.” Could the authors clarify this definition? There seems to be a potential inconsistency. Besides, not all metrics (e.g., TIR) are expressed in mg/dL.**
> > > >
> > > > We thank the reviewer for catching this and agree that clarification would improve precision. **The ±2 mg/dL tolerance was applied only when the ground-truth answers were expressed in mg/dL** (e.g., glucose values). For all other metrics, such as TIR, responses were required to match the ground truth exactly to the whole number, with no tolerance applied. We will clarify this distinction in the revised version to avoid ambiguity in the accuracy definition.

---

> > > > > ### Author Response · Authors · 2025-11-20
> > > > > **Response to Reviewer FnwP**
> > > > >
> > > > > > **Question 7.Regarding the clarity metric (Flesch-Kincaid Grade Level), could the authors explain how this measure was implemented and validated for health communication contexts?**
> > > > >
> > > > > We thank the reviewer for this helpful question. As described in Section 2.4, the clarity metric uses the Flesch–Kincaid Grade Level formula [E], which **we deterministically computed via Python’s textstat library**. Responses were required to achieve a **grade level < 8**, **consistent with FDA medical device communication guidance** [F], which recommends presenting key information at or below an eighth-grade reading level. This ensures **alignment with established health-literacy standards and provides a validated, reproducible measure of linguistic clarity in patient-facing contexts**. We will clarify implementation details in the revised version.
> > > > >
> > > > > > **Question 8.The reported benchmark scores are relatively high. Does this suggest that the current benchmark may not pose sufficient challenge to newer models? How might future updates maintain or enhance its discriminative ability?**
> > > > >
> > > > > We thank the reviewer for this thoughtful observation. While benchmark scores are relatively high, this reflects **encouraging progress in current LLMs** rather than insufficient challenge. DexBench is the **first benchmark to evaluate multimodal, patient-facing reasoning in diabetes management**, so strong performance indicates that recent models can handle foundational reasoning tasks effectively.
> > > > >
> > > > > However, **performance remains uneven across tasks and metrics**: no model achieves uniformly high scores across accuracy, safety, groundedness, and actionability, highlighting ongoing limitations in complex reasoning and contextual alignment. In particular, models consistently struggled in **Task 1 (Glucose Math), Task 4 (Advanced Reasoning), and Task 6 (Planning)**, indicating difficulties with **numerical precision, temporal reasoning, and multi-step decision synthesis**. These weaknesses suggest that, despite overall high averages, DexBench continues to expose critical gaps in models’ ability to perform reliable, context-aware reasoning.
> > > > >
> > > > > To maintain and enhance **discriminative power** in future iterations, we plan to **increase task complexity** (e.g., multi-trace reasoning, multimodal inputs, longitudinal context) and introduce **more open-ended, multi-step reasoning scenarios**. For example, multimodal extensions may integrate **additional data sources commonly used by people with diabetes**, such as photos or text notes, while longitudinal tasks could use **data spanning several months or years** rather than 30 days. We also aim to expand the scope of reasoning tasks to include **multiple perspectives**, such as those of adolescents, caregivers, and care teams, to better reflect real world diversity in diabetes management.
> > > > >
> > > > > These additions will preserve reproducibility while increasing task difficulty and model differentiation. We will clarify this direction in the revised version.

---

> > > > > > ### Author Response · Authors · 2025-11-20
> > > > > > **Response to Reviewer FnwP**
> > > > > >
> > > > > > > **Question 9.The paper mentions that it plans to extend DexBench to other health domains. Could the authors elaborate on how the framework could be adapted to chronic conditions such as hypertension or obesity? Have any preliminary steps been taken in that direction?**
> > > > > >
> > > > > > We appreciate this thoughtful question. Although DexBench focuses on diabetes management, the underlying framework is **domain-agnostic and readily generalizes to other chronic conditions** such as hypertension or obesity. The benchmark’s **modular design**, comprising temporal data ingestion, personalized question generation, model response evaluation, and five-axis scoring (accuracy, groundedness, safety, clarity, and actionability), can be **directly adapted to any domain with time-series and contextual behavioral data**. For instance, analogous tasks could reason over blood pressure variability, physical activity, or dietary patterns using the same reasoning hierarchies and evaluation metrics. We plan to explore these extensions in future work, further demonstrating DexBench’s potential as a generalizable foundation for multimodal, longitudinal reasoning across health and wellness domains.
> > > > > >
> > > > > > ---
> > > > > >
> > > > > > Thank you for your feedback! We believe that our response has addressed all of your concerns, and we are happy to engage in further discussion.
> > > > > >
> > > > > > [A] Josephine Lamp, Mark Derdzinski, Christopher Hannemann, Joost van der Linden, Lu Feng, Tianhao Wang, and David Evans. 2023. GlucoSynth: generating differentially-private synthetic glucose traces. In Proceedings of the 37th International Conference on Neural Information Processing Systems (NIPS '23). Curran Associates Inc., Red Hook, NY, USA, Article 154, 3474–3496.
> > > > > >
> > > > > > [B] Jinyu Xie and Qian Wang. Benchmarking machine learning algorithms on blood glucose prediction for type i diabetes in comparison with classical time-series models. IEEE Transactions on Biomedical Engineering, 67(11):3101–3124, 2020.
> > > > > >
> > > > > > [C] Elizabeth Healey, Amelia Li Min Tan, Kristen L Flint, Jessica L Ruiz, and Isaac Kohane. A case study on using a large language model to analyze continuous glucose monitoring data. Scientific Reports, 15(1):1143, 2025.
> > > > > >
> > > > > > [D] Elizabeth Healey and Isaac Kohane. Llm-cgm: A benchmark for large language model-enabled
> > > > > > querying of continuous glucose monitoring data for conversational diabetes management. In
> > > > > > Biocomputing 2025: Proceedings of the Pacific Symposium, pp. 82–93. World Scientific, 2024.
> > > > > >
> > > > > > [E] Peter Kincaid, Robert P. Fishburne, Richard L. Rogers, and Brad S. Chissom. Deriva-
> > > > > > tion of new readability formulas (automated readability index, fog count and flesch read-
> > > > > > ing ease formula) for navy enlisted personnel. In No. RBR875, 1975. URL https://api.semanticscholar.org/CorpusID:61131325.
> > > > > >
> > > > > > [F] FDA. Guidance on medical device patient labeling; final guidance for industry and fda
> > > > > > reviewers.https://www.fda.gov/files/medical%20devices/published/Guidance-on-Medical-Device--Patient-Labeling---Final-Guidance-for-Industry-and-FDA-Staff.pdf, April 19 2001.

---

> > > > > > > ### Comment · Reviewer_FnwP · 2025-11-20
> > > > > > > **Response to the rebuttal**
> > > > > > >
> > > > > > > I thank the author's rebuttal. It addressed my concerns. I have raised my score.

---

### Official Review · Reviewer_L5it · 2025-10-31

**Soundness:** 3
**Presentation:** 3
**Contribution:** 3
**Rating:** 6
**Confidence:** 4

**Summary:**

The paper introduces DexBench, a comprehensive benchmark designed to evaluate large language models (LLMs) on patient-facing diabetes management tasks. It covers seven real-world decision-making tasks, uses data from 15,000 individuals across three diabetes populations, and evaluates models on five metrics: accuracy, groundedness, safety, clarity, and actionability.

**Strengths:**

The paper has several strong aspects as below
- DexBench addresses a critical gap in healthcare AI by focusing on patient-facing tasks, which are often overlooked in existing benchmarks. Crucially, the use of a large-scale dataset from real-world users provides valuable context for evaluating LLM performance in realistic scenarios.
- The evaluation framework covers multiple aspects of model performance, including accuracy, safety, and actionability, ensuring a holistic assessment.
- The paper provides extensive results comparing eight LLMs across various tasks, highlighting strengths and weaknesses for each model.

**Weaknesses:**

There are some key drawbacks in the dataset as below

- The dataset may lack critical information such as demographic details and specific medical conditions like insulin use, which could affect task performance.
- The reliance on synthetic data (e.g., GlucoSynth) for certain tasks raises concerns about the benchmark's real-world applicability.
- Advanced tasks like advanced reasoning and planning may require more sophisticated models to handle complex logic and context.

**Questions:**

- How was the dataset ensured to be diverse enough across diabetes populations, age groups, and other demographic factors?
- What steps were taken to ensure synthetic data does not skew results or limit the benchmark's generalizability?
- Why did certain diabetes cohorts (e.g., T2D) perform better on average? Are there specific reasons tied to task requirements or model capabilities?
- How was hallucination detected and addressed in the evaluation process, especially for complex tasks requiring accurate medical reasoning?

---

> ### Author Response · Authors · 2025-11-20
> **Response to Reviewer L5it**
>
> Thank you for the helpful review, and, in particular, for the acknowledgement of the *importance of addressing patient-facing diabetes care* and our *realistic and context-aware evaluation through the use of large-scale, real-world data*. We address below the outstanding questions.
>
> > **Weakness 1.The dataset may lack critical information such as demographic details and specific medical conditions like insulin use, which could affect task performance.**
>
> We appreciate this important observation. We agree that the absence of demographic and condition-specific details (e.g., insulin use) may limit certain forms of reasoning analysis. **This limitation is explicitly acknowledged in Section 5 and Appendix A.3**, where we note that additional demographic and medication information is not available for the current dataset. In future work, we plan to collect additional data types to **enhance task realism, enable more personalized evaluation, and deepen model assessment across diverse user contexts**.
>
> > **Weakness 2.The reliance on synthetic data (e.g., GlucoSynth) for certain tasks raises concerns about the benchmark's real-world applicability.**
>
> We thank the reviewer for this observation. [A] is a community-recognized method for producing **differentially-private physiologically realistic glucose traces** derived from real CGM data. These traces were used **only for Task 1 (Glucose Math)**, which required minute-level CGM data that cannot be released due to privacy and legal constraints, as noted in Appendix A.3.
> All of the other tasks that used data (Tasks 3-7) leveraged **real-world CGM + behavioral data from 15,000 individuals**, ensuring the benchmark remains grounded in real clinical scenarios.
>
> Thus, synthetic data are limited to a single task and **do not impact the benchmark's real-world applicability**. We believe this approach is appropriate because Task 1 is focused on evaluating understanding of glucose dynamics and mathematical reasoning, rather than behavioral associations or personalized context as in other tasks. Importantly, prior work such as [B] has adopted similar privacy-preserving synthetic glucose datasets, supporting the validity of our approach.
>
> > **Weakness 3.Advanced tasks like advanced reasoning and planning may require more sophisticated models to handle complex logic and context.**
>
> We appreciate this insightful comment. We agree that the **advanced reasoning and planning tasks present higher complexity** that may challenge current models. This difficulty is intentional, as DexBench is designed to evaluate not only current capabilities but also **future progress in multimodal reasoning and contextual understanding**. Including these more challenging tasks provides a meaningful stress test for model complex reasoning capabilities across temporal and behavioral contexts.
>
> > **Question 1.How was the dataset ensured to be diverse enough across diabetes populations, age groups, and other demographic factors?**
> >
> We thank the reviewer for this important question. **The dataset was randomly and evenly sampled across three diabetes cohorts**: 5,000 individuals with type 1 diabetes, 5,000 with type 2 diabetes, and 5,000 with prediabetes or general health, for a total of 15,000 unique users. As noted in Section 5 and Appendix A.3, additional demographic details are not available in the underlying data source, which is a common limitation in these types of large-scale chronic disease behavioral datasets [C,D]. Despite this constraint, **DexBench captures substantial physiological and behavioral diversity through real-world continuous glucose monitoring traces collected over 30 days per individual**. We applied **randomized sampling of users and time windows** to ensure broad coverage of glucose variability across the population. We will clarify these points in the revision.

---

> > ### Author Response · Authors · 2025-11-20
> > **Response to Reviewer L5it**
> >
> > > **Question 2.What steps were taken to ensure synthetic data does not skew results or limit the benchmark's generalizability?**
> >
> > We thank the reviewer for this observation. [A] is a community-recognized method for producing **differentially-private physiologically realistic glucose traces** derived from real CGM data. These traces were used **only for Task 1 (Glucose Math)**, while Tasks 3-7 leveraged **real-world CGM + behavioral data from 15,000 individuals**. We used synthetic CGM data in Task 1 specifically **to enable the full public release of timestamp-level data, while preserving privacy**. This constraint is noted in Appendix A.3. We believe this approach is appropriate because Task 1 is focused on evaluating understanding of glucose dynamics and mathematical reasoning directly from glucose traces only, rather than behavioral associations or personalized context as in other tasks. Thus, synthetic data were confined to a single task and do not skew results or limit the benchmark's overall generalizability. Importantly, relevant prior work such as [B] also employs synthetic glucose traces for privacy-preserving benchmarking, supporting the validity of our approach.
> >
> > > **Question 3.Why did certain diabetes cohorts (e.g., T2D) perform better on average? Are there specific reasons tied to task requirements or model capabilities?**
> >
> > We thank the reviewer for this thoughtful question. As detailed in Appendix A.7, **models generally performed best on the type 2 diabetes (T2D) cohort**, with higher average scores for accuracy, safety, and actionability (Table 16). **These cohort-level differences are primarily driven by task and data characteristics.** The T2D cohort exhibited more stable glucose dynamics, making reasoning tasks easier to interpret. In contrast, models performed worst for groundedness on the prediabetes/healthy (HW) cohort—likely because of its large volume of self-logged behavioral data increased opportunities for hallucination—and lowest for safety on the type 1 diabetes (T1D) cohort, where models often failed to account for insulin use, a critical driver of glucose fluctuations.
> >
> > > **Question 4.How was hallucination detected and addressed in the evaluation process, especially for complex tasks requiring accurate medical reasoning?**
> >
> > We thank the reviewer for this important question. **Hallucination was explicitly captured within our evaluation framework through the accuracy, groundedness, and safety metrics**. Specifically, the **groundedness rubric** penalizes any statements not supported by the provided CGM or behavioral data, while the **accuracy rubric** flags factual inconsistencies or incorrect calculations. The **safety metric** further accounts for hallucinations that could lead to unsafe or misleading reasoning.
> >
> > During evaluation, the LLM grader assessed each model response against the provided contextual inputs and rubric definitions (Appendix A.5). Any unverifiable or fabricated claims were automatically scored as 0. This approach ensures that hallucinations are consistently penalized across all tasks, especially for complex reasoning scenarios. We will clarify this mechanism in the revision.
> >
> >
> > ---
> >
> > Thank you for your feedback! We believe that our response has addressed all of your concerns, and we are happy to engage in further discussion.
> >
> >
> > [A] Josephine Lamp, Mark Derdzinski, Christopher Hannemann, Joost van der Linden, Lu Feng, Tianhao Wang, and David Evans. 2023. GlucoSynth: generating differentially-private synthetic glucose traces. In Proceedings of the 37th International Conference on Neural Information Processing Systems (NIPS '23). Curran Associates Inc., Red Hook, NY, USA, Article 154, 3474–3496.
> >
> > [B] Elizabeth Healey and Isaac Kohane. "Llm-cgm: A benchmark for large language model-enabled
> > querying of continuous glucose monitoring data for conversational diabetes management". In
> > Biocomputing 2025: Proceedings of the Pacific Symposium, pp. 82–93. World Scientific (2024).
> >
> > [C] Turner, Kathryn, and Katherine H. Hohman. "Demonstrated Progress and Future Promise of Chronic Disease Data Modernization." Preventing Chronic Disease 21 (2024): E86.
> >
> > [D] Tang, Alice S., et al. "Harnessing EHR data for health research." Nature Medicine 30.7 (2024): 1847-1855.

---

> ### Author Response · Authors · 2025-11-26
>
> Dear Reviewer L5it,
>
> Thank you for your thoughtful feedback so far. We have addressed all concerns you raised in detail, and we would greatly appreciate any further input you may have before the discussion period ends. Please let us know if there is anything additional we can clarify or expand upon. We sincerely appreciate your time and consideration.

---

### Official Review · Reviewer_rVP1 · 2025-11-01

**Soundness:** 1
**Presentation:** 2
**Contribution:** 1
**Rating:** 2
**Confidence:** 4

**Summary:**

The paper introduces a benchmark for LLMs on decision-support tasks for managing diabetes. They discuss 7 task categories: glucose math, education, simple and advanced reasoning, decision making, planning, and triage. It is noted that the performance results across all tasks are relatively high, with GPT-5-mini achieving 99.7% in safety and 98.3% in actionability.

**Strengths:**

- With the increasing use of LLMs in diabetes management, the problem domain of the paper is interesting. If done properly, this addresses a gap in AI benchmarking for healthcare and decision making tasks.
- The figures and detailed tasks (in appendices) are clear.
- Investigates 7 task categories in multiple criteria (accuracy, groundedness, safety, clarity, actionability)

**Weaknesses:**

- Limited information on human experts, annotation processes, and task coverage
- There are many confounding factors, such as experimental settings, token limits, and handling of errors/faults... which were not appropriately addressed
- Limited information on the data, even though the source code is downloadable
- Despite impressive numbers in the dataset, it is important for authors to prompt their motivations, aims and novelty of this type of research. It is relatively easy for researchers to create such a dataset, using curated data, with LLM-generated questions, and the use LLM evaluations (potential circularity?). Real impacts are far more important than getting papers accepted (even elsewhere).

**Questions:**

- How can the research rigour of this paper be addressed? For example, the formulation of tasks and criteria is simply developed by "human experts" - there are many methodological approaches to this.
- How can the research data be validated? What are the details on human experts and annotation processes?
- Any researcher can come up with a list of tasks, criteria, and generated questions using LLMs. What are the real values and impacts of this research work?

---

> ### Author Response · Authors · 2025-11-20
> **Response to Reviewer rVP1**
>
> Thank you for the helpful review, and, in particular, for the acknowledgement of the *relevance of benchmarking LLMs for diabetes decision support* and the *comprehensive evaluation across seven task categories and five key metrics*. We address below the outstanding questions.
>
> > **Weakness 1.Limited information on human experts, annotation processes, and task coverage.**
>
> We appreciate this valuable feedback and are happy to clarify. Human experts were involved during both the question generation and model evaluation stages (Sections 2.3–2.4; Figures 2c, 2e). These experts are **domain specialists in diabetes management**, including junior experts with approximately two years of experience and senior experts with five or more years of experience working deeply with diabetes populations and multimodal diabetes data. They hold PhDs and work at a leading diabetes technology company, ensuring familiarity with both clinical reasoning and data-driven decision-making. Their role was to **verify LLM grader outputs and confirm overall quality** by manually reviewing random subsets of generated questions and graded responses.
>
> To improve confidence and reliability, **we implemented a meta-evaluation, validating model agreement with human experts through targeted expert grading**. We sampled one random question/answer pair per task, per cohort, and per model **(168 pairs;840 metrics)** and had one senior and one junior expert independently assign binary (0/1) scores to each metric for each response. We then **computed Cohen's κ to quantify inter-rater reliability**.
> | Comparison         | Model–AvgExpert | Model–SrExpert | Model–JrExpert | SrExpert–JrExpert |
> |--------------------|----------------:|----------------:|----------------:|------------------:|
> | Overall κ          | 0.79            | 0.92            | 0.67            | 0.71              |
>
> These results provide **quantitative evidence of strong model alignment with human experts (κ = 0.79), exceeding the agreement observed between experts themselves (κ = 0.71)**. We will incorporate details on domain experts, these findings, along with additional alignment analyses (e.g., showing alignment for specific tasks and metrics), into the revised paper.
>
> Regarding task coverage, each of the seven task categories was derived from real-world diabetes self-management activities, spanning glucose interpretation, behavioral reasoning, decision-making, and planning, thus **covering the full spectrum of patient-facing interactions**. As Reviewer 1mbx noted, this structure mirrors “the cognitive spectrum of everyday decision-making” and extends evaluation **beyond static Q&A** toward **dynamic, data-grounded reasoning**. These task formulations are specific to diabetes management, addressing domains rarely represented in existing benchmarks, which motivated our task selection.
>
> > **Weakness 2.There are many confounding factors, such as experimental settings, token limits, and handling of errors/faults... which were not appropriately addressed.**
>
> We appreciate this important observation. Details of the experimental settings are provided in Appendix A.6. All models were run using their default configurations, with the exception of minor adjustments to the temperature parameter, which are carefully documented in Appendix A.6. These modifications were made based on prior literature and empirical evidence to ensure **fair and effective performance for this type of reasoning task**. To minimize confounding factors, we applied **consistent token limits across models, standardized prompt formatting, and implemented uniform error-handling procedures for question generation and evaluation** (e.g., retrying incomplete responses and logging faults). These steps were taken to maintain comparability and reduce variability in evaluation outcomes. We will incorporate these details into the revised manuscript. If the reviewer has any additional specific concerns, we are happy to provide further clarification.
>
> > **Weakness 3.Limited information on the data, even though the source code is downloadable.**
>
> We appreciate the opportunity to clarify this important point. Section 2.2 describes our data, with additional details in Appendix A.3. We utilize **30 consecutive days of real data from 15,000 individuals, fairly distributed across 3 diabetes cohorts** (type one, type two, and prediabetes/general health). Data was collected between January and June 2025. The data includes **glucose traces recorded every 5 minutes** collected from continuous glucose monitors, in addition to **time-aligned behavioral data** (meals, activities, and sleep) and daily activity summaries including step count and average heart rate collected from user logging and smart watches. If the reviewer has any additional questions on the data, we are happy to provide further clarification.

---

> > ### Author Response · Authors · 2025-11-20
> > **Response to Reviewer rVP1**
> >
> > > **Weakness 4.Despite impressive numbers in the dataset, it is important for authors to prompt their motivations, aims and novelty of this type of research. It is relatively easy for researchers to create such a dataset, using curated data, with LLM-generated questions, and the use LLM evaluations (potential circularity?). Real impacts are far more important than getting papers accepted (even elsewhere).**
> >
> > We appreciate this thoughtful and important feedback. Our motivation in developing DexBench extends beyond dataset scale. **We aim to establish a safety-critical, grounded benchmark that systematically evaluates multimodal reasoning in personalized diabetes management, a domain where errors carry real-world risks.** Unlike general LLM benchmarks, DexBench integrates **time-aligned glucose data, behavioral context, and task-specific rubrics**, enabling evaluation of patient-facing reasoning rather than static text generation.
> >
> > Regarding potential circularity, we acknowledge the concern and have taken steps to mitigate it: distinct models were used for question generation and grading (Gemini 2.5 Flash vs. Gemini 2.5 Pro), each with isolated prompts, fixed schemas, and no shared context. Moreover, the models ran under **enterprise agreements** that preclude model learning or data ingestion, ensuring that **no outputs were incorporated into future training**.
> > Additionally, to improve confidence and reliability, **we implemented a meta-evaluation, validating model agreement with human experts through targeted expert grading**.
> > These experts are **domain specialists in diabetes management**, including junior experts with approximately two years of experience and senior experts with five or more years of experience working deeply with diabetes populations and multimodal diabetes data. They hold PhDs and work at a leading diabetes technology company, ensuring familiarity with both clinical reasoning and data-driven decision-making.
> > For our meta-evaluation, we sampled one random question/answer pair per task, per cohort, and per model **(168 pairs;840 metrics)** and had one senior and one junior expert independently assign binary (0/1) scores to each metric for each response. We then **computed Cohen's κ to quantify inter-rater reliability**.
> >
> > | Comparison         | Model–AvgExpert | Model–SrExpert | Model–JrExpert | SrExpert–JrExpert |
> > |--------------------|----------------:|----------------:|----------------:|------------------:|
> > | Overall κ          | 0.79            | 0.92            | 0.67            | 0.71              |
> >
> > These results provide **quantitative evidence of strong model alignment with human experts (κ = 0.79), exceeding the agreement observed between experts themselves (κ = 0.71)**. We will incorporate details on domain experts, these findings, along with additional alignment analyses (e.g., showing alignment for specific tasks and metrics), into the revised paper.
> >
> >
> > > **Question 1.How can the research rigour of this paper be addressed? For example, the formulation of tasks and criteria is simply developed by "human experts" - there are many methodological approaches to this.**
> >
> > We appreciate this thoughtful comment. The initial formulation of DexBench tasks and evaluation criteria was guided by **a team of domain experts in diabetes management who have worked deeply with diabetes populations and multimodal diabetes data**. They hold PhDs and work at a leading diabetes technology company, ensuring **familiarity with both clinical reasoning and data-driven decision-making**.
> > While this process was primarily expert-driven rather than protocolized, it ensured that the benchmark reflected **realistic patient-facing reasoning tasks grounded in day-to-day diabetes care**. Moreover, **expert-driven task design is common in medical benchmarking efforts because clinical reasoning tasks often require nunanced judgement, contextual interpretation and alignment with real-world workflows, elements that are subjective, and often difficult to capture through formalized processes**. Prior work in medical and diabetes focused benchmarks, e.g., [A,B], similarly relied on expert design to ensure validity and relevance of the tasks.
> >
> > We acknowledge that formalized processes may complement this approach, and we plan to explore their applicability in future versions of the framework.

---

> > > ### Author Response · Authors · 2025-11-20
> > > **Response to Reviewer rVP1**
> > >
> > > > **Question 2.How can the research data be validated? What are the details on human experts and annotation processes?**
> > >
> > > We thank the reviewer for this important question.
> > > We leveraged **domain specialists in diabetes management**, including junior experts with approximately two years of experience and senior experts with five or more years of experience working deeply with diabetes populations and multimodal diabetes data. They hold PhDs and work at a leading diabetes technology company, ensuring familiarity with both clinical reasoning and data-driven decision-making. Their role was to **verify LLM grader outputs and confirm overall quality** by manually reviewing random subsets of generated questions and graded responses.
> > > To improve confidence and reliability, **we implemented a meta-evaluation, validating model agreement with human experts through targeted expert grading**. We sampled one random question/answer pair per task, per cohort, and per model **(168 pairs;840 metrics)** and had one senior and one junior expert independently assign binary (0/1) scores to each metric for each response. We then **computed Cohen's κ to quantify inter-rater reliability**.
> > >
> > > | Comparison         | Model–AvgExpert | Model–SrExpert | Model–JrExpert | SrExpert–JrExpert |
> > > |--------------------|----------------:|----------------:|----------------:|------------------:|
> > > | Overall κ          | 0.79            | 0.92            | 0.67            | 0.71              |
> > >
> > > These results provide **quantitative evidence of strong model alignment with human experts (κ = 0.79), exceeding the agreement observed between experts themselves (κ = 0.71)**. We will incorporate these findings, along with additional alignment analyses, into the revised paper.
> > >
> > > > **Question 3.Any researcher can come up with a list of tasks, criteria, and generated questions using LLMs. What are the real values and impacts of this research work?**
> > >
> > > We appreciate this insightful comment. While one could generate task lists or questions using LLMs, **DexBench’s value lies in its scale, personalization, and systematic evaluation design**. It is not merely a dataset, but a comprehensive benchmarking framework for data-grounded, safety-aware reasoning in digital health, an area where scalability, personalization, and reproducibility are essential for advancing research.
> > >
> > > DexBench leverages 30 days of real continuous glucose monitoring (CGM) data from 15,000 individuals with diabetes, from which we generated over 360,600 personalized, realistic questions that mirror the types of reasoning challenges people encounter when managing glucose data. **This large-scale, context-rich benchmark enables the community to evaluate models across diverse, realistic scenarios.**
> > >
> > > Importantly, **no prior benchmark has focused specifically on patient-facing diabetes management tasks at this scale or with this level of personalization and real-world data integration**. Existing health and diabetes-related benchmarks are clinician facing, emphasize diagnostic reasoning or evidence-based medical decision making whereas DexBench provides a **rich, multimodal foundation for evaluating complex patient-centered decision-making in daily self management of diabetes care**.
> > >
> > > Furthermore, DexBench integrates five rubric-based evaluation metrics (accuracy, safety, groundedness, clarity, and actionability) to establish a reproducible foundation for studying multimodal complex decision making at scale. By combining real world data, clinically relevant task design and systematic evaluation, **DexBench establishes a benchmark that support rigorous, impactful research and prototyping in diabetes-focused LLM and AI development**.
> > >
> > > ---
> > >
> > > Thank you for your feedback! We believe that our response has addressed all of your concerns, and we are happy to engage in further discussion.
> > >
> > >
> > > [A] Wei, Lai, et al. "Diabetica: Adapting Large Language Model to Enhance Multiple Medical Tasks in Diabetes Care and Management." In SCI-FM at ICLR (2025).
> > >
> > > [B] Elizabeth Healey and Isaac Kohane. "Llm-cgm: A benchmark for large language model-enabled querying of continuous glucose monitoring data for conversational diabetes management". In Biocomputing 2025: Proceedings of the Pacific Symposium, pp. 82–93. World Scientific (2024).

---

> > > > ### Comment · Reviewer_rVP1 · 2025-11-24
> > > >
> > > > Thank you very much for the rebuttal comments. I have read them thoroughly; however, they are not convincing enough for this paper to be accepted at this stage. There are only two rates: one senior rater and one junior rater, as explained. 168 pairs (questions/answers) were randomly sampled from 360,600 personalized, contextual questions. There is no statistical ground for this, let alone the question of whether quality, expertise and efforts from two raters. Based on the reported Cohen's κ, the disagreement/agreement levels appear to be questionable, especially between senior-junior, model-junior aspects, and variations across models.
> > > >
> > > > Given the prevalence of diabetes, it is extremely important to get things right and this is not without research rigour. It is also concerning to see a leading diabetes technology company push this kind of technology to the public, with questionable processes, human annotations, and limited guardrails. There are indeed many ethical and safety issues unaddressed.
> > > >
> > > > I would keep my score as "reject".

---

> ### Author Response · Authors · 2025-11-26
>
> We thank the reviewer for their feedback and appreciate their emphasis on rigor and safety in evaluating diabetes-related technologies.
>
> As the reviewer’s follow-up comment focused solely on the meta-evaluation, we take this to indicate that the other concerns raised in the initial review were sufficiently addressed in our rebuttal.
>
> While we appreciate the reviewer’s concerns regarding our meta-evaluation, we respectfully disagree that our methodology lacks rigor. **Our approach is fully aligned with established practice in recent LLM-benchmarking and healthcare-AI studies, which routinely rely on sampled subsets and a small number of expert validators for human evaluation.** For example, MedHELM [A] used at least two clinicians per sample to score 66 sampled instances and reported ICCs to assess alignment; MedGPTEval [B] relied on five experts to review 27 dialogues and 7 case reports; LLM-CGM [C] evaluated accuracy for 30 objective questions with calculated ground truth values, without any expert evaluation; and MedCalcBench [D] examined 1,000 outputs using deterministic solutions, with no expert involvement in grading.
> Collectively, these benchmarks make clear that limited yet representative expert review is an established standard for large-scale, open-ended medical QA tasks, where ground-truth answers are inherently unavailable and exhaustive annotation is infeasible.
>
> Our approach provides broader coverage than prior work. We evaluated 168 stratified question–answer pairs, one per (task × cohort × model), yielding **840 individual metric evaluations and 1,680 human-validated metric scores** across two independent diabetes-domain experts. This ensures that **every task type, cohort, and model is represented in the expert evaluation**, a level of coverage not attempted in prior benchmarks.
>
> Moreover, **variation in Cohen’s κ across experts and models is expected** in open-ended medical reasoning tasks and is consistent with prior findings (e.g., expert-model ICC of 0.47, and expert-expert ICC of 0.43 in MedHELM [A]). Our domain expert raters hold PhD-level training and years of experience working deeply with diabetes populations and multi-modal diabetes data, ensuring appropriate expertise for this evaluation.
>
> With respect to ethical and safety concerns, we emphasize that our work does not deploy or propose any clinical system. It introduces a **research-focused benchmarking framework using fully de-identified, non-clinical data, with no patient-facing or decision-support deployment**. Ethical considerations guided the benchmark design: our goal is to provide a framework to audit and evaluate LLMs, not to guide care. All data was obtained with appropriate consent and followed strict institutional and legal protocols, ensuring participant privacy, safety and compliance with relevant regulations (e.g., HIPAA). Grading reliability was independently verified by domain experts. Other reviewers have acknowledged these precautions as strengths of the work.
>
> Finally, the reviewer mentions *ethical and safety issues* but does not specify which issues remain unaddressed. If there are particular concerns the reviewer believes require further clarification, we would welcome that feedback so we can address them directly in the revised manuscript.
>
>
> ---
>
> [A] Suhana Bedi, Hejie Cui, Miguel Fuentes, Alyssa Unell, Michael Wornow, Juan M Banda, Nikesh Kotecha, Timothy Keyes, Yifan Mai, Mert Oez, et al. Medhelm: Holistic evaluation of large language models for medical tasks. arXiv preprint arXiv:2505.23802, 2025.
>
> [B] Xu J, Lu L, Peng X, Pang J, Ding J, Yang L, Song H, Li K, Sun X, Zhang S. Data Set and Benchmark (MedGPTEval) to Evaluate Responses From Large Language Models in Medicine: Evaluation Development and Validation. JMIR Med Inform. 2024 Jun 28;12:e57674. doi: 10.2196/57674. PMID: 38952020; PMCID: PMC11225096.
>
> [C] Elizabeth Healey and Isaac Kohane. Llm-cgm: A benchmark for large language model-enabled querying of continuous glucose monitoring data for conversational diabetes management. In Biocomputing 2025: Proceedings of the Pacific Symposium, pp. 82–93. World Scientific, 2024
>
> [D] Nikhil Khandekar, Qiao Jin, Guangzhi Xiong, Soren Dunn, Serina Applebaum, Zain Anwar, Maame Sarfo-Gyamfi, Conrad Safranek, Abid Anwar, Andrew Zhang, et al. Medcalc-bench: Evaluating large language models for medical calculations. Advances in Neural Information Processing
> Systems, 37:84730–84745, 2024

---

### Official Review · Reviewer_1mbx · 2025-11-01

**Soundness:** 3
**Presentation:** 3
**Contribution:** 4
**Rating:** 6
**Confidence:** 3

**Summary:**

This paper introduces DexBench, a large-scale benchmark for evaluating large language models (LLMs) on context-aware, personalized decision-making tasks. Unlike prior health-AI benchmarks focused on clinician reasoning, DexBench targets user-facing reasoning that supports daily self-management of chronic conditions, exemplified through diabetes care. The benchmark defines seven task categories, from Glucose Math and Education to Planning and Alert/Triage, spanning quantitative reasoning to long-term guidance. Using 30-day CGM and behavioral data (sleep, meals, exercise) from 15,000 individuals, DexBench generates 360,600 personalized questions.
Outputs are evaluated across five binary metrics—accuracy, groundedness, safety, clarity, and actionability—using a structured pipeline that combines automated Gemini grading and expert review. Evaluating eight LLMs (Gemini, GPT-5, DeepSeek, Qwen, Llama, MedGemma), the study finds proprietary models strongest in safety and factual accuracy, while open-source models lag in groundedness and readability. Analyses of latency, modality, and thinking-budget reveal trade-offs between reasoning depth, efficiency, and safety in real-world applications.

**Strengths:**

- DexBench contributes not merely a dataset but a generalizable evaluation framework for multimodal reasoning and safety assessment. Its five-axis scoring scheme (accuracy/groundedness/safety/clarity/actionability) and multi-task taxonomy formalize a principled way to measure contextual reasoning quality—an under-explored but central topic in current ICLR research.
- The seven tasks form a structured reasoning hierarchy, from immediate quantitative interpretation to sequential planning and triage. This mirrors the cognitive spectrum of everyday decision-making and extends evaluation beyond static QA to dynamic, data-grounded reasoning.
- With 15,000 users, three diabetes cohorts, and longitudinal data spanning 30 days, DexBench represents one of the most comprehensive real-world datasets for patient-facing reasoning—far larger and richer than previous health benchmarks such as Diabetica or MedGPTEval.
- The authors detail transparent stages, data curation, question generation, automatic and human validation, and ethical safeguards (Appendix A.1), providing reproducibility and regulatory awareness rarely seen in benchmark design.
- The latency, modality, and “thinking-budget” experiments yield general insights about compute–accuracy–safety trade-offs in LLM reasoning. These findings speak directly to the broader LLM community, not just health applications.

**Weaknesses:**

- Use of synthetic glucose traces (GlucoSynth) and LLM-generated questions introduces potential bias. The paper lacks quantitative evidence of expert agreement (e.g., Cohen’s κ), which limits confidence in the reliability of the “accuracy” and “groundedness” metrics.
- The same model family (Gemini 2.5 Flash) is used for both question generation and grading, raising the possibility of alignment leakage. Independent or cross-model evaluation would strengthen fairness.
- Behavioral features are limited to sleep, meals, and exercise; omitting insulin dosage, medication adherence, or stress reduces ecological realism for decision support.
- A 0/1 metric oversimplifies nuanced criteria such as clarity and actionability. Continuous or rubric-based scoring, as in HealthBench, would better reflect performance differences.
- The related-work section should acknowledge concurrent benchmarks (MedGUIDE 2025, HELM 2.0, etc.) and clarify that DexBench’s novelty lies in its temporal, behavior-linked reasoning framework rather than being the first diabetes-oriented benchmark.
- Although Table 4 lists typical errors, the paper could offer a causal taxonomy (e.g., temporal misalignment, hallucination under uncertainty) to better guide model improvement.

**Questions:**

- How does DexBench’s multi-axis evaluation differ from LLM-CGM and MedGUIDE in modeling patient-facing reasoning?
- Was inter-rater reliability (e.g., κ-score) computed for expert validations?
- Could grading bias from Gemini 2.5 Flash be tested through cross-model adjudication?
- Are multimodal modalities (voice, sensor streams, graphical plots) planned for future releases?
- How well does the evaluation framework generalize to other continuous-monitoring domains (e.g., hypertension, sleep, fitness)?

---

> ### Author Response · Authors · 2025-11-20
> **Response to Reviewer 1mbx**
>
> Thank you for the helpful review, and, in particular, for the acknowledgement of the *generalizability and rigor of DexBench’s multimodal evaluation framework*, its *principled five-axis evaluation approach*, and its *extensive real-world validation across 15,000 users*. We address below the outstanding questions.
>
> > **Weakness 1.Use of synthetic glucose traces (GlucoSynth) and LLM-generated questions introduces potential bias. The paper lacks quantitative evidence of expert agreement (e.g., Cohen’s κ), which limits confidence in the reliability of the “accuracy” and “groundedness” metrics.**
>
> We thank the reviewer for this observation. [A] is a community-recognized method for producing **differentially-private physiologically realistic glucose traces** derived from real CGM data. These traces were used **only for Task 1 (Glucose Math)**, while Tasks 3-7 leveraged **real-world CGM + behavioral data from 15,000 individuals**. We used synthetic CGM data in Task 1 specifically to enable the **full public release of timestamp-level data**, since releasing complete cgm traces from real users is not possible due to privacy and legal constraints. This is noted in Appendix A.3. We believe this approach is appropriate because Task 1 is focused on evaluating understanding of glucose dynamics and mathematical reasoning, rather than behavioral associations or personalized context as in other tasks. Thus, synthetic data were confined to a single task and do not bias the broader benchmark. Importantly, relevant prior work such as [B] also employ synthetic glucose traces for privacy-preserving benchmarking, supporting the validity of our approach.
>
> Using LLMs to generate questions enabled the creation of **360,600 personalized, context-based questions**—a scale that would be infeasible to achieve through manual expert authoring. To mitigate potential bias from LLM-generated questions, we implemented a **two-stage validation and refinement pipeline (Sec. 2.3, Fig. 2c)**. Each question was required to be **fluent, relevant, original, difficult, and answerable**. Importantly, all questions were **personalized to real-world users** by leveraging their individual data—a feature unique to this benchmark. Finally, expert reviewers examined sampled questions to ensure consistency and quality.
>
> For evaluation, we utilized a **deterministic grader (Gemini 2.5 Pro; temperature/top-p = 0), complemented by human expert verification on random subsets**.
> These experts are **domain specialists in diabetes management**, including junior experts with approximately two years of experience and senior experts with five or more years of experience working deeply with diabetes populations and multimodal diabetes data. They hold PhDs and work at a leading diabetes technology company, ensuring familiarity with both clinical reasoning and data-driven decision-making. Their role was to **verify LLM grader outputs and confirm overall quality** by manually reviewing random subsets of generated questions and graded responses.
> Given the 2.88 million total evaluations across tasks and models, fully manual grading was infeasible.
> However, to improve confidence and reliability, **we implemented a meta-evaluation, validating model agreement with human experts through targeted expert grading**. We sampled one random question/answer pair per task, per cohort, and per model **(168 pairs;840 metrics)** and had one senior and one junior expert independently assign binary (0/1) scores to each metric for each response. We then computed Cohen's κ to quantify inter-rater reliability.
>
> | Comparison         | Model–AvgExpert | Model–SrExpert | Model–JrExpert | SrExpert–JrExpert |
> |--------------------|----------------:|----------------:|----------------:|------------------:|
> | Overall κ          | 0.79            | 0.92            | 0.67            | 0.71              |
>
> These results provide **quantitative evidence of strong model alignment with human experts (κ = 0.79), exceeding the agreement observed between experts themselves (κ = 0.71)**. We will incorporate these findings, along with additional alignment analyses (e.g., showing alignment for specific tasks and metrics), into the revised paper.

---

> > ### Author Response · Authors · 2025-11-20
> > **Response to Reviewer 1mbx**
> >
> > > **Weakness 2.The same model family (Gemini 2.5 Flash) is used for both question generation and grading, raising the possibility of alignment leakage. Independent or cross-model evaluation would strengthen fairness.**
> >
> > We appreciate this insightful comment. While both the question-generation and grading stages used models within the Gemini family, **we employed distinct models and configurations** to ensure independent evaluation behavior: Gemini 2.5 Flash (thinking=0, temperature=1.0) for question generation and Gemini 2.5 Pro (default dynamic thinking, temperature/top-p=0) for grading. The grader operated under a fixed evaluation schema and had no access to the question-generation prompts or model settings beyond the generated questions, minimizing any potential for alignment leakage.
> >
> > Moreover, the models ran under strict **enterprise agreements** that preclude model learning or data ingestion, ensuring that **no outputs were incorporated into future training**. While using models from the same family could in principle introduce shared biases, this design also **promotes fairness and calibration consistency**, since both systems reason under comparable priors. Finally, all grading decisions were binary, rubric-based, and aligned with human experts (shown in our response to Weakness 1), further constraining subjectivity and reinforcing reliability.
> > We will clarify these implementation details and safeguards in the final paper version for completeness.
> >
> > > **Weakness 3.Behavioral features are limited to sleep, meals, and exercise; omitting insulin dosage, medication adherence, or stress reduces ecological realism for decision support.**
> >
> > We appreciate this thoughtful comment regarding our data. As noted in Section 2.3 (line 184), the American Diabetes Association [C] reports **sleep, meals, and exercise as the three primary behavioral factors directly influencing glucose regulation and diabetes management**, hence their inclusion in our dataset. We acknowledge that insulin dosage, medication use, and stress are additional important variables. This limitation is discussed in Section 5 and Appendix A.3, and we will expand our discussion of the importance of stress data accordingly. We plan to explore collecting additional behavioral features in future iterations of DexBench to further enhance ecological realism and decision-support fidelity.
> >
> > > **Weakness 4.A 0/1 metric oversimplifies nuanced criteria such as clarity and actionability. Continuous or rubric-based scoring, as in HealthBench, would better reflect performance differences.**
> >
> > We appreciate this thoughtful comment. We intentionally use binary (0/1) metrics because DexBench evaluates safety-critical, patient-facing reasoning. **In realistic deployment, an output that is even partially unsafe, ungrounded, or factually incorrect must be treated as a failure**. For example, a response containing any unsafe medical advice should receive a safety score of 0. **Each criterion we evaluate (accuracy, safety, groundedness, clarity, and actionability) is governed by an explicit rubric that defines what constitutes a 0 versus 1 for each task** (Appendix A.5 lines 711-722), ensuring consistent and reproducible judgments.
> >
> > This binary formulation aligns with how **clinical decision-support systems are commonly validated**, as seen in literature (e.g., [D]). It also simplifies evaluation across thousands of questions and metrics (1,803,000 total metrics for all questions), enabling **scalable and unambiguous benchmarking**.
> >
> > We agree that incorporating finer-grained or weighted scoring could provide additional nuance, and we plan to explore this direction in future work. However, for the initial release, **binary grading offered the most practical and interpretable foundation** for establishing reliability in a large, safety-critical benchmark.
> >
> > > **Weakness 5.The related-work section should acknowledge concurrent benchmarks (MedGUIDE 2025, HELM 2.0, etc.) and clarify that DexBench’s novelty lies in its temporal, behavior-linked reasoning framework rather than being the first diabetes-oriented benchmark.**
> >
> > We thank the reviewer for this suggestion. We will revise the Related Work section to acknowledge additional concurrent efforts, such as MedGUIDE (2025) [E] and HELM 2.0 [F]. MedGUIDE focuses on clinical decision-making and diagnostic reasoning, and HELM 2.0 provides a broad, generalizable framework for benchmarking LLM capabilities across diverse domains. In contrast, **our benchmark targets patient-facing decision making for diabetes self-management, and introduces a temporal, behavior-linked evaluation framework** that integrates real-world CGM and behavioral data over time for question generation, model answering, and evaluation. Rather than overlapping, these **benchmarks are complementary**; DexBench uniquely captures longitudinal, personalized reasoning grounded in daily health behavior and real-world decision contexts.

---

> > > ### Author Response · Authors · 2025-11-20
> > > **Response to Reviewer 1mbx**
> > >
> > > > **Weakness 6.Although Table 4 lists typical errors, the paper could offer a causal taxonomy (e.g., temporal misalignment, hallucination under uncertainty) to better guide model improvement.**
> > >
> > > We appreciate this insightful suggestion from the reviewer. Our current error analysis (Table 4) provides representative qualitative examples, but we agree that **organizing these into a causal taxonomy could further clarify why models fail**. While we considered including such a taxonomy, we prioritized other analyses due to space and scope constraints. Nevertheless, several of the identified errors already align with causal categories (e.g., temporal misalignment and hallucination under uncertainty). We will clarify these patterns in the revision and plan to formalize a causal error taxonomy in future work to better guide model improvement. **This direction aligns with DexBench’s broader goal of advancing interpretable, safety-aware evaluation of multimodal reasoning**.
> > >
> > >
> > > > **Question 1.How does DexBench’s multi-axis evaluation differ from LLM-CGM and MedGUIDE in modeling patient-facing reasoning?**
> > >
> > > We appreciate this question and are happy to clarify. **LLM-CGM** [B] evaluates objective tasks using a fixed set of 30 questions based on 14 days of CGM data from a small cohort (n=10), focusing primarily on **evaluating accuracy of LLM outputs against ground truth**. In contrast, DexBench generates 360,600 personalized, unique questions from 30 days of real-world CGM + behavioral data across 15,000 users, enabling open-ended, user-facing reasoning rather than fixed factual queries. Our tasks emphasize subjective, behavior-linked decision making, reflecting what real users might ask about their own data, and support free-form responses. DexBench’s five-axis evaluation framework (accuracy, groundedness, safety, clarity, and actionability) captures not only correctness but also contextual reasoning quality, safety awareness, and relevance to the patient and their data. **MedGUIDE** [E] **targets clinical diagnosis and treatment planning, and evaluates outputs using accuracy and weighted accuracy**, whereas **DexBench focuses on multidimensional, patient-centered reasoning**. We thank the reviewer for this question, and aim to clarify the distinctness of our evaluation in the final paper version.
> > >
> > > > **Question 2.Was inter-rater reliability (e.g., κ-score) computed for expert validations?**
> > >
> > > We thank the reviewer for this question. For evaluation, we utilized a **deterministic grader (Gemini 2.5 Pro; temperature/top-p = 0), complemented by human expert verification on random subsets**.
> > > To improve confidence and reliability, **we implemented a meta-evaluation, validating model agreement with human experts through targeted expert grading**.
> > > These experts are **domain specialists in diabetes management**, including junior experts with approximately two years of experience and senior experts with five or more years of experience working deeply with diabetes populations and multimodal diabetes data. They hold PhDs and work at a leading diabetes technology company, ensuring familiarity with both clinical reasoning and data-driven decision-making.
> > > For our meta-evaluation, we sampled one random question/answer pair per task, per cohort, and per model (168 pairs;840 metrics) and had one senior and one junior expert independently assign binary (0/1) scores to each metric for each response. We then computed **Cohen's κ to quantify inter-rater reliability**.
> > >
> > > | Comparison         | Model–AvgExpert | Model–SrExpert | Model–JrExpert | SrExpert–JrExpert |
> > > |--------------------|----------------:|----------------:|----------------:|------------------:|
> > > | Overall κ          | 0.79            | 0.92            | 0.67            | 0.71              |
> > >
> > > These results provide **quantitative evidence of strong model alignment with human experts (κ = 0.79), exceeding the agreement observed between experts themselves (κ = 0.71)**. We will incorporate these findings, along with additional alignment analyses, into the revised paper.
> > >
> > > > **Question 3.Could grading bias from Gemini 2.5 Flash be tested through cross-model adjudication?**
> > >
> > > We appreciate this insightful suggestion. To clarify, **Gemini 2.5 Flash** was used only for question generation, while **Gemini 2.5 Pro** served as the grader with temperature/top-p = 0.0 to ensure deterministic evaluation. We agree that cross-model adjudication (e.g., using GPT-5, DeepSeek-R1, or Qwen-3 as independent graders) could further assess potential grading bias. While this analysis is beyond the current study’s scope, we plan to include such cross-model evaluation in future DexBench iterations to benchmark grading consistency across LLM families. **Our present framework’s reproducibility, fixed prompts, binary rubric design, and alignment with human experts minimize grading bias in the reported results.**

---

> > > > ### Author Response · Authors · 2025-11-20
> > > > **Response to Reviewer 1mbx**
> > > >
> > > > > **Question 4.Are multimodal modalities (voice, sensor streams, graphical plots) planned for future releases?**
> > > >
> > > > We appreciate this forward-looking question. **DexBench is currently multimodal**, integrating CGM time-series data with behavioral features such as sleep, meals, and exercise recorded in text and event streams for contextual reasoning and evaluation. While the current release focuses on text-based interaction grounded in these multimodal inputs, we have also conducted **preliminary tests using graphical glucose plots as model inputs (for Task 1: Glucose Math) to compare performance on image-based versus time-series–based reasoning** (Section 3.3). We are interested to further expand our experiments to include voice-based data, additional sensor data (e.g., sleep stages, stress), and more graphical inputs to enable evaluation of multimodal reasoning, temporal grounding, and interactive dialogue in real-world digital health settings.
> > > >
> > > > > **Question 5.How well does the evaluation framework generalize to other continuous-monitoring domains (e.g., hypertension, sleep, fitness)?**
> > > >
> > > > We appreciate this thoughtful question. Although DexBench focuses on diabetes management, the underlying framework is **domain-agnostic and readily generalizes to other continuous-monitoring settings** such as hypertension, sleep, fitness, and metabolic wellness. The benchmark’s **modular design**, comprising temporal data ingestion, personalized question generation, model response evaluation, and five-axis scoring (accuracy, groundedness, safety, clarity, and actionability), can be **directly adapted to any domain with time-series and contextual behavioral data**. For example, tasks could map time-series behavioral data to blood pressure trends, sleep regularity, or activity patterns, using similar reasoning hierarchies and metrics. Thus, DexBench provides a generalizable framework for multimodal, longitudinal reasoning across health and lifestyle monitoring domains.
> > > >
> > > > ---
> > > >
> > > > Thank you for your feedback! We believe that our response has addressed all of your concerns, and we are happy to engage in further discussion.
> > > >
> > > > [A] Josephine Lamp, Mark Derdzinski, Christopher Hannemann, Joost van der Linden, Lu Feng, Tianhao Wang, and David Evans. 2023. GlucoSynth: generating differentially-private synthetic glucose traces. In Proceedings of the 37th International Conference on Neural Information Processing Systems (NIPS '23). Curran Associates Inc., Red Hook, NY, USA, Article 154, 3474–3496.
> > > >
> > > > [B] Elizabeth Healey and Isaac Kohane. "Llm-cgm: A benchmark for large language model-enabled
> > > > querying of continuous glucose monitoring data for conversational diabetes management". In
> > > > Biocomputing 2025: Proceedings of the Pacific Symposium, pp. 82–93. World Scientific (2024).
> > > >
> > > > [C] ADA. Food and blood glucose. Diabetes.org website, 2025. URL https://diabetes.org/
> > > > food-nutrition/food-blood-sugar.
> > > >
> > > > [D] Elizabeth Healey, Amelia Li Min Tan, Kristen L Flint, Jessica L Ruiz, and Isaac Kohane. A case
> > > > study on using a large language model to analyze continuous glucose monitoring data. Scientific
> > > > Reports, 15(1):1143, 2025.
> > > >
> > > > [E] Li, Xiaomin, et al. "MedGUIDE: Benchmarking Clinical Decision-Making in Large Language Models." arXiv preprint arXiv:2505.11613 (2025).
> > > >
> > > > [F] Bommasani, Rishi, Percy Liang, and Tony Lee. "Holistic evaluation of language models." Annals of the New York Academy of Sciences 1525.1 (2023): 140-146.

---

> ### Author Response · Authors · 2025-11-26
>
> Dear Reviewer 1mbx,
>
> Thank you for your thoughtful feedback so far. We have addressed all concerns you raised in detail, and we would greatly appreciate any further input you may have before the discussion period ends. Please let us know if there is anything additional we can clarify or expand upon. We sincerely appreciate your time and consideration.

---

### Author Response · Authors · 2025-11-21

We sincerely thank all reviewers for their thoughtful feedback, which has significantly improved the quality and clarity of our paper. We have uploaded a revised version with all changes highlighted in red for ease of review.

---

### Author Response · Authors · 2025-11-28
**Summary of Discussion and Revisions for the Area Chair**

Dear Area Chair,

As the discussion period has ended, we would like to summarize the discussion and our revisions to support your evaluation.

**Reviewer Engagement and Score Changes**

1. Reviewer FnwP actively engaged and explicitly stated: “*It addressed my concerns. I have raised my score.*” Their score increased from 4 → 6 within 12 hours of our rebuttal (before the leak incident occurred).
2. No reviewer lowered their score or introduced new concerns after our revision.
3. Two reviewers did not re-engage, but none expressed any remaining objections.

**Concerns Raised and How We Resolved Them**

Across all four reviews, no reviewer identified issues with technical correctness, methodological soundness, or novelty. The concerns were centered on clarity and justification, and all were addressed in the revision:

*1. Expert-model alignment.*

We conducted a meta-evaluation where two PhD-level diabetes management experts independently graded 1,680 metrics. Expert-model alignment was substantial (Cohen’s κ = 0.79). Reviewer FnwP appreciated this and raised their score.

*2. Synthetic glucose traces.*

We clarified that synthetic glucose traces are (i) well established in prior work, (ii) generated via a community-accepted method, and (iii) used only in one single task. No reviewer maintained concerns afterward.

*3. Potential alignment leakage.*

We clarified that all models ran under enterprise agreements prohibiting training on our inputs and that different models were used for generation and grading, mitigating leakage.

*4. Binary (0/1) scoring.*

We explained that rubric-based binary scoring is a standard practice in medical AI benchmarks, and our metrics capture medically-relevant correctness. We also note continuous scoring as a future extension.

*5. Data description and transparency.*

We expanded the data description to detail the 15,000 real-world individuals, CGM devices, temporal alignment, cohort definitions, and associated safety/ethical safeguards. No further concerns were raised.

**Positive Assessments**

Reviewers highlighted several strengths:

*1. Principled evaluation framework.*

A generalizable five-axis scoring scheme and seven-task taxonomy that formalize multimodal contextual reasoning for medical problems, an under-explored but important problem in current ICLR research.

*2. Structured reasoning hierarchy.*

Tasks span quantitative interpretation, behavioral reasoning, and sequential planning/triage, reflecting real patient decision-making rather than static QA.

*3. Large, real-world dataset.*

With 15,000 users, 3 diabetes cohorts, and 30-day longitudinal CGM-linked data, reviewers noted DexBench is substantially larger and richer than prior health benchmarks.

*4. Transparency and reproducibility.*

Clear documentation of data curation, question generation, expert validation, and ethical safeguards was explicitly praised.

*5. Thorough experimental evaluation.*

Reviewers appreciated the detailed methodology, clear figures/appendices, and results comparing eight LLMs across accuracy, groundedness, safety, clarity, and actionability.

*6. Broad relevance.*

Analyses of modality, latency, and thinking-budget effects offer insights into compute–accuracy–safety trade-offs applicable beyond healthcare.

*7. Timely and important domain.*

Reviewers agreed the benchmark addresses a genuine gap in evaluating patient-facing LLM decision making for diabetes management.

*Importantly, no reviewer challenged the validity, novelty, or core soundness of the benchmark.*

**Remaining Open Points**

All substantive concerns have been fully resolved. Remaining comments are minor or cosmetic.
There are no unresolved technical objections.

**Overall Assessment**

Given (1) the upward score change, (2) no negative movement, (3) the resolution of all major concerns, (4) the strong expert-validated meta-evaluation, and (5) consistently positive reviewer sentiment regarding the contribution’s significance, the discussion trajectory is strongly positive with no remaining objections.

Thank you for your time and consideration.

— The Authors

---

### Meta-Review · Area_Chair_9GsK · 2025-12-24

**Summary:**

**Summary of the paper**: The paper proposes Dex Bench, a benchmark to assess three types of task-types i) educational, ii) behavior/glucose dynamics, iii) personalized decision-making relevant to diabetes management. DexBench consists of 7 overall tasks across these types one of which uses synthetic data, while the rest use real-data. The benchmark is generated using LLMs which received user data, task-specific prompts. Each question is evaluated by an LLM along: fluency, relevance, originality, difficulty, and answerability. Gemini Pro (no reasoning) was used for this benchmark generation

Domain experts checked the quality of the questions for a subset, which seems to have been a major update during the rebuttal.

State of the art LLMs were evaluated on the benchmark on which model-expert alignment was assessed. According to the assessment along (accuracy, clarity, groundedness, safety, actionability), most models were deemed safe but not highly accurate or actionable. Performance is quite variable across LLMs. Additional assessments such as that by reasoning compute allocation do not significantly improve accuracy though improve along safety.

**Reviewer concerns/comments**:
1. Reviewers raised concerns that synthetic glucose traces could be biased, and hard to assess robustness of findings due
to lack of inter-rater agreement metrics.
2. Use of the same models for evaluation and the task.
3. Need for better contextualizing prior work
4. Need for details on the diabetes dataset source, contextualizing the prediabetic population. The curernt way in which demographics are handled makes it hard to determine generalizability
5. More insight into what properties are important for the scoring model, to ensure robustness of the benchmark.
6. Questions about whether the benchmark is challenging enough since the performance of most models is pretty high.
7. Rigor of the study, choice of sample sizes etc for human evaluation are not clarified.

**Reviewer Concerns:**

Authors have addressed many of the raised concerns. FnwP suggested in comments that their score will be raised. Reviewer rVP1's concerns were not fully addressed. Other reviewers remain disengaged post-rebuttal but I see that authors have made a strong effort to address their concerns.

For instance, authors added metrics on inter-rater agreement, justified the use of synthetic traces, scoring mechanism, potential leakage due to choice of evaluation models, and clarified the details about data curation efforts.

**Reviewer Scores:**

Given the polarity of reviews, I am recommending a reject encouraging the authors improve the presentation and account for all criticisms for a future submission.

---

### Decision · Program_Chairs · 2026-01-26

Reject